# Examining Inter-Consistency of Large Language Models Collaboration: An In-depth Analysis via Debate

**Kai Xiong**[1]   **Xiao Ding**[1†]   **Yixin Cao**[2†]   **Ting Liu**[1]   **Bing Qin**[1]

[1]Research Center for Social Computing and Information Retrieval
Harbin Institute of Technology, China
[2]Singapore Management University, Singapore

{kxiong, xding, tliu, qinb}@ir.hit.edu.cn
yxcao@smu.edu.sg

## Abstract

Large Language Models (LLMs) have shown impressive capabilities in various applications, but they still face various inconsistency issues. Existing works primarily focus on the inconsistency issues within a single LLM, while we complementarily explore the inter-consistency among multiple LLMs for collaboration. To examine whether LLMs can collaborate effectively to achieve a consensus for a shared goal, we focus on commonsense reasoning, and introduce a formal debate framework (FORD) to conduct a three-stage debate among LLMs with real-world scenarios alignment: fair debate, mismatched debate, and roundtable debate. Through extensive experiments on various datasets, LLMs can effectively collaborate to reach a consensus despite noticeable inter-inconsistencies, but imbalances in their abilities can lead to domination by superior LLMs. Leveraging a more advanced LLM like GPT-4 as an authoritative judge can boost collaboration performance. Our work contributes to understanding the inter-consistency among LLMs and lays the foundation for developing future collaboration methods. Codes and data are available at https://github.com/Waste-Wood/FORD.

## 1 Introduction

Large Language Models (LLMs) like ChatGPT recently demonstrate general intelligence (Bubeck et al., 2023) and have been widely used as a foundation model in various applications (Wei et al., 2022b; Wu et al., 2023). To solve complex tasks, multiple LLMs are further introduced to collaborate, with each targeting a different subtask or aspect (Schick et al., 2022; Park et al., 2023). Interestingly, do these LLMs possess a spirit of collaboration? Are they capable of cooperating effectively and performantly towards a shared goal?

In this paper, we dive into the inter-consistency among LLMs, complementary to existing work

---

[†]Corresponding Authors

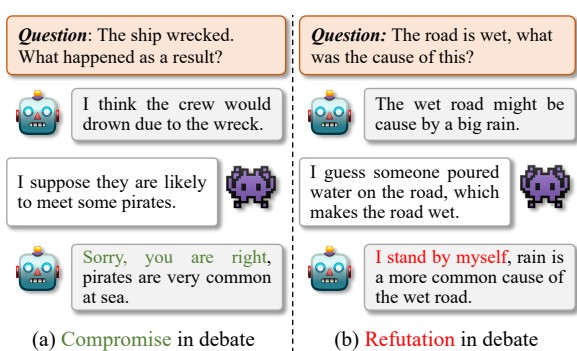

Question: The ship wrecked. What happened as a result?

I think the crew would drown due to the wreck.

I suppose they are likely to meet some pirates.

Sorry, you are right, pirates are very common at sea.

Question: The road is wet, what was the cause of this?

The wet road might be cause by a big rain.

I guess someone poured water on the road, which makes the road wet.

I stand by myself, rain is a more common cause of the wet road.

(a) Compromise in debate    (b) Refutation in debate

Figure 1: (a) compromise and (b) refutation in LLMs debates. 🤖 is the proponent, and 👾 is the opponent.

which mostly investigates the self-consistency issue of a single LLM (Wang et al., 2022b; Jung et al., 2022). Based on the observations, we highlight two main inter-consistent concerns for LLMs' collaboration. First, the viewpoints of LLMs are easily shifted. As shown in Figure 1 (a), the proponent and opponent LLMs are showing different predictions, while the proponent quickly compromises with the answer of the opponent. To which extent do LLMs easily change their viewpoints, or adhere to their perspectives? Second, when some LLMs remain steadfast in Figure 1 (b), can a consensus ultimately be achieved for the shared goal?

Inspired by the debate theory (Mayer, 1997), we devise a **For**mal **D**ebate framework (FORD) to systematically and quantitatively investigate the inter-inconsistency issue in LLMs collaboration. Based on FORD, we allow LLMs to explore the differences between their own understandings and the conceptualizations of others via debate (Mayer, 1997). Thus, the results can not only shed light to encourage more diverse results, but also make it possible for performance gains by mutual learning.

In specific, we take multi-choice commonsense reasoning as the example task as it can accurately quantify the inter-inconsistency of LLMs collaboration. Then we formulate a three-stage debate to align with real-world scenarios: (1) **Fair de-**

**bate** between two LLMs with comparable capabilities. (2) **Mismatched debate** between two LLMs who exhibit vastly different levels of abilities. (3) **Roundtable debate** including more than two LLMs for debates. Besides, due to the leading performance of GPT-4 (OpenAI, 2023), we conduct an in-depth analysis to adopt GPT-4 as a more powerful judge to summarize the debates and offer final conclusions. Note that FORD is flexible if stronger or more LLMs emerge.

We summarize our main findings as follows. (1) Different types of LLMs (e.g., chat and text completion models) have large inter-inconsistency, even if they are developed from the same base model. (2) For different versions of an LLM within the same series (e.g., GPT-3.5), although their overall performance keeps improving, the more advanced models do not completely supersede the capabilities of the early ones. (3) Through FORD, multiple LLMs hold the potential to collaborate and reach a consensus, while the final results are not always so desirable. (4) For LLMs with comparable abilities, they can effectively and performantly achieve a shared goal. (5) On the other hand, for LLMs with mismatched abilities, the superior LLMs are more likely to insist on their perspectives and dominate the debate, while weaker ones are more likely to compromise and change their viewpoints. (6) Nevertheless, the stubborn and less capable LLMs may distract the superior ones and lower the overall performance.

We summarize our contributions as follows:

- We systematically and quantitatively study the inter-consistency among two and more LLMs.

- We adopt debate theory and design a formal debate framework FORD towards quantitative analysis and LLMs collaboration.

- We design a three-stage debate and have conducted extensive experiments to offer insights for future research.

## 2 Preliminaries

We first describe the datasets and LLMs used in this paper. Then we define a metric INCON to quantify the inter-inconsistency among multiple LLMs. Finally, we define the baselines to verify our proposed formal debate framework FORD.

| Dataset | Task Type | Size |
|---|---|---|
| $\alpha$**NLI** (Bhagavatula et al., 2019) | 2 Choices | 1,507 |
| **CSQA** (Talmor et al., 2019) | 5 Choices | 1,221 |
| **COPA** (Gordon et al., 2012) | 2 Choices | 500 |
| **e-CARE** (Du et al., 2022a) | 2 Choices | 2,122 |
| **Social IQa** (Sap et al., 2019) | 3 Choices | 1,935 |
| **PIQA** (Bisk et al., 2020) | 2 Choices | 1,838 |
| **StrategyQA** (Geva et al., 2021) | Yes or No | 2,290 |

Table 1: Commonsense reasoning dataset statistics.

### 2.1 Commonsense Reasoning Datasets

For better coverage, we choose 7 multi-choice commonsense reasoning datasets for experiments: one abductive reasoning dataset $\alpha$NLI (Bhagavatula et al., 2019), one commonsense question answering dataset CSQA (Talmor et al., 2019), two causal reasoning datasets COPA (Gordon et al., 2012) and e-CARE (Du et al., 2022a), one social interaction reasoning dataset Social IQa (Sap et al., 2019), one physical interaction question answering dataset PIQA (Bisk et al., 2020), and one implicit reasoning strategy dataset StrategyQA (Geva et al., 2021). Table 1 shows the statistics of the above datasets.

### 2.2 Large Language Models

We choose 6 LLMs from multiple sources for experiments. We first choose 4 LLMs from OpenAI: three chat completion LLMs `gpt-3.5-turbo` (denoted as ChatGPT), `gpt-3.5-turbo-0301` (denoted as ChatGPT-0301), and `gpt-4` (denoted as GPT-4), as well as one text completion LLM `text-davinci-003` (denoted as Davinci-003). Then we adopt two open-source 13B LLMs: an efficient foundation LLM LLaMA (Touvron et al., 2023), and Vicuna (Chiang et al., 2023) which is trained on 70K data from ShareGPT.

### 2.3 Definitions: `INCON`

Here we define the metric `INCON` to quantify the inter-inconsistency among multiple LLMs. Specifically, suppose there are $n$ LLMs $L = \{l_1, \cdots, l_n\}$, and a dataset with $m$ samples $X = \{x_1, \cdots, x_m\}$. We define $p_j^i$ as the prediction of $l_i$ on $x_j$. Then the `INCON` of $L$ on $X$ can be defined as:

$$\text{INCON} = \sum_{k=1}^{m} \frac{\Phi(p_k^1, \cdots, p_k^n)}{m}, \qquad (1)$$

$\Phi$ is a sign function, it will be assigned a value of 1 if there are any two variables in $\Phi$ that are not equal, otherwise, $\Phi$ takes a value of 0.

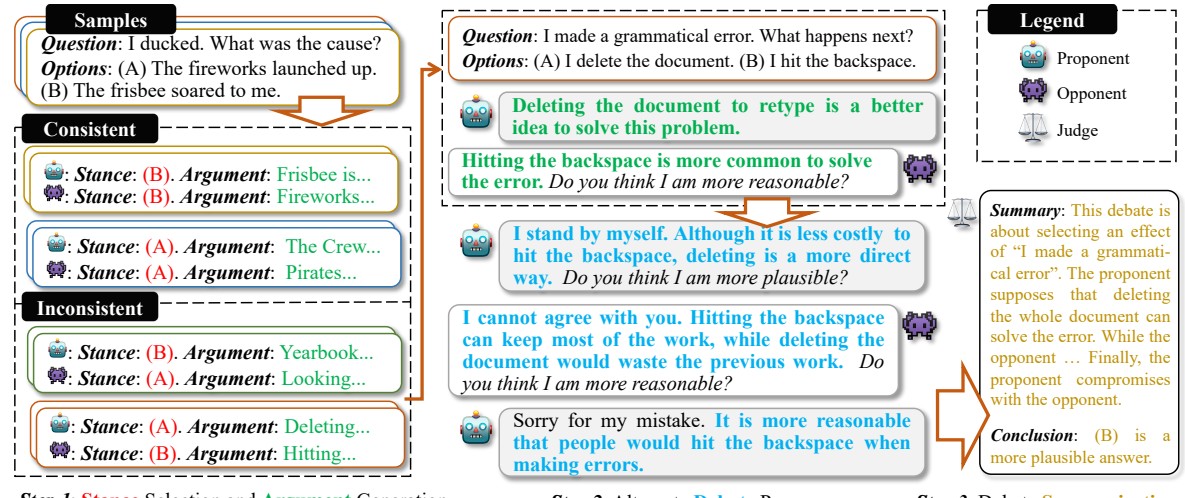

**Step 1**: **Stance** Selection and **Argument** Generation     **Step 2**: Alternate **Debate** Process     **Step 3**: Debate **Summarization**

Figure 2: The overall workflow of FORD on two LLMs. **Step 1**: LLMs independently give a choice and explanation for each sample as the stance and argument, respectively. **Step 2**: LLMs debate alternately on the inconsistent samples. **Step 3**: A judge summarizes the whole debate process and gives the final conclusion.

## 2.4 Baseline Methods

We define three kinds of baselines to verify our proposed formal debate framework: (1) **Single LLM** use only one LLM to conduct experiments. (2) **Collaboration-Soft** (`Col-S`) aligns the loose scenario that we randomly trust one of the LLMs when they are inconsistent, so `Col-S` averages the accuracy of the LLMs. (3) **Collaboration-Hard** (`Col-H`) aligns the conservative scenario that we only trust consistent prediction, so `Col-H` predicts correctly when all LLMs predict correctly.

## 3 FORD: Formal Debate Framework

To further explore the inter-inconsistency, inspired by the structure of debate (Bell, 1998), we design a formal debate framework (FORD) for LLMs collaboration to solve shared tasks. In Figure 2, we use two LLMs for illustration, FORD consists of 3 steps: (1) **Stance selection and argument generation** expects LLMs to choose a stance and give an argument for each sample, aiming to select the inconsistent samples for future collaboration in step 2. (2) **Alternate debate process** makes LLMs alternately refute each other or seek a compromise to investigate whether LLMs can collaborate effectively towards a consensus. (3) **Debate summarization** uses a judge to summarize the debate, then draw a final conclusion in case does not reach a consensus.

### 3.1 Stance Selection & Argument Generation

The first step of FORD is to choose a stance, then provide an argument to support the stance. Hence,

we ask each LLM to independently choose a choice and a short explanation as the stance and argument, respectively. Experimental details and prompts can refer to Appendix A. As shown in step 1 of Figure 2, when asked about the cause of "*I ducked*", the proponent would hold the stance "*(B)*" and gives the argument "*Frisbee is usually a potential threat to safety*" to support its stance.

We focus on the inconsistent samples in the subsequent steps to investigate LLMs collaboration.

### 3.2 Alternate Debate Process

To investigate whether LLMs can effectively collaborate and achieve a consensus, we design an alternate debate process. Given the arguments in step 1, the proponent conducts round 1 debate to first counter the opponent by presenting an updated argument or to seek a compromise. If no compromise, the opponent would conduct round 2 to do the same to the proponent by additionally considering the arguments produced in previous rounds. The debate process will be conducted alternately until the LLMs achieve a consensus, or the debate rounds reach the maximum number. To make LLMs less swing, the stance will not explicitly show in step 2. Prompts and details can refer to Appendix B.

As shown in step 2 in Figure 2, the proponent defends itself by claiming "*deleting is a more direct way*", while the opponent adheres to its perspective by stating "*hitting backspace*" is less costly. Finally, the proponent compromise with the opponent.

| Debate | Method | αNLI | CSQA | COPA | e-CARE | Social IQa | PIQA | StrategyQA | *Average* |
|---|---|---|---|---|---|---|---|---|---|
| ChatGPT & Davinci-003 | ChatGPT | 77.17 | 76.17 | 96.80 | 81.53 | 72.71 | 80.74 | 69.34 | 79.21 |
| | Davinci-003 | 79.96 | 78.62 | 95.20 | 79.69 | 73.33 | 76.88 | 71.48 | 79.31 |
| | Col-S | 78.75 | 77.40 | 96.00 | 80.61 | 73.02 | 78.81 | 70.41 | 79.29 |
| | Col-H | 69.14 | 68.22 | 93.80 | 71.96 | 62.69 | 68.55 | 58.12 | 70.35 |
| | FORD (Ours) | 81.35 | 78.79 | 97.00 | 83.74 | 74.32 | 84.60 | 71.62 | 81.63 |
| ChatGPT & ChatGPT-0301 | ChatGPT | 77.17 | 76.17 | 96.80 | 81.53 | 72.71 | 80.74 | 69.34 | 79.21 |
| | ChatGPT-0301 | 77.70 | 75.43 | 97.20 | 81.53 | 73.54 | 80.79 | 69.17 | 79.34 |
| | Col-S | 77.44 | 75.80 | 97.00 | 81.53 | 73.13 | 80.77 | 69.26 | 79.28 |
| | Col-H | 74.78 | 73.46 | 96.00 | 79.55 | 70.96 | 77.97 | 67.25 | 77.14 |
| | FORD (Ours) | 78.30 | 76.41 | 97.60 | 81.95 | 73.75 | 82.70 | 69.65 | 80.05 |
| LLaMA & Vicuna | LLaMA | 63.04 | 66.18 | 90.20 | 66.97 | 58.98 | 71.60 | 63.10 | 68.58 |
| | Vicuna | 70.60 | 59.87 | 89.20 | 68.05 | **64.65** | 60.45 | 55.28 | 66.87 |
| | Col-S | 66.82 | 63.02 | 89.70 | 67.51 | 61.82 | 66.03 | 59.15 | 67.72 |
| | Col-H | 54.74 | 47.91 | 83.40 | 53.72 | 46.77 | 48.26 | 41.35 | 53.74 |
| | FORD (Ours) | 70.80 | 66.75 | 93.80 | 71.54 | 63.67 | 72.25 | 64.10 | 71.84 |

Table 2: The overall performance of FORD and baselines in the fair debates. Underlined numbers represent the best performance among the collaboration methods on each dataset. **Bold** numbers denote the best performance among all methods on each dataset. "Average" denotes the mean accuracy across different datasets for each method.

### 3.3 Debate Summarization

To obtain the final result of the debate, we adopt a judge to summarize the whole debate and draw a conclusion. We design a template to obtain the summary by filling it with the arguments, which can provide an overview of the debate process to enhance the interpretability. The conclusion is about the result of the debate. For samples that reached a consensus, the conclusion is the consensus stance. For samples without a consensus, we assign equal weights to all arguments for the conclusion. Refer to Appendix E for more details.

Next, based on the above framework, we simulate three debate scenarios for comprehensive investigation: (1) **Fair debate** between two LLMs with comparable capabilities (Sec 4). (2) **Mismatched debate** between two LLMs with mismatched capabilities (Sec 5). (3) **Roundtable debate** including more than two LLMs for debate (Sec 6).

### 4 Fair Debate

Given multi-choice questions, we conduct fair debates on three pairs of LLMs: different types of LLMs (ChatGPT & Davinci-003 and LLaMA & Vicuna), and different versions of LLMs (ChatGPT & ChatGPT-0301). $\text{LLM}_P$ & $\text{LLM}_O$ denotes $\text{LLM}_P$ as the proponent, and the $\text{LLM}_O$ is the opponent.

### 4.1 Initial INCON of LLMs Pairs

We first conduct step 1 on each LLM on each dataset. The temperature is set as 0 for reproducibil-

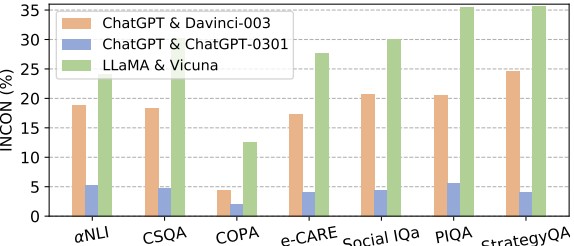

Figure 3: Initial INCON of LLMs pairs on each dataset. The dashed part in each bar denotes the INCON brought by the wrong predictions of the proponent.

ity. The initial INCON of each LLMs pair on each dataset is categorized in Figure 3:

(1) Different types of LLMs hold nearly 20%-30% INCON on almost all datasets, even if they are based on the same underlying model. The overlapped part in each bar contributes nearly 50% to INCON, which means LLMs in each LLMs pair possess comparable yet vastly different capabilities.

(2) For ChatGPT & ChatGPT-0301, ChatGPT-0301 does not supersede ChatGPT in capabilities. This indicates LLMs gain new abilities as they iterate but lose some existing ones. Hence, it is not convincing to use the updated LLMs to reproduce the results of the unavailable early versions.

### 4.2 Results of Fair Debate

Through steps 2 & 3 in FORD, we intervene the predictions under the setting of fair debate. Heuristically, for ChatGPT & Davinci-003 and LLaMA & Vicuna, the maximum rounds of debate is 6.

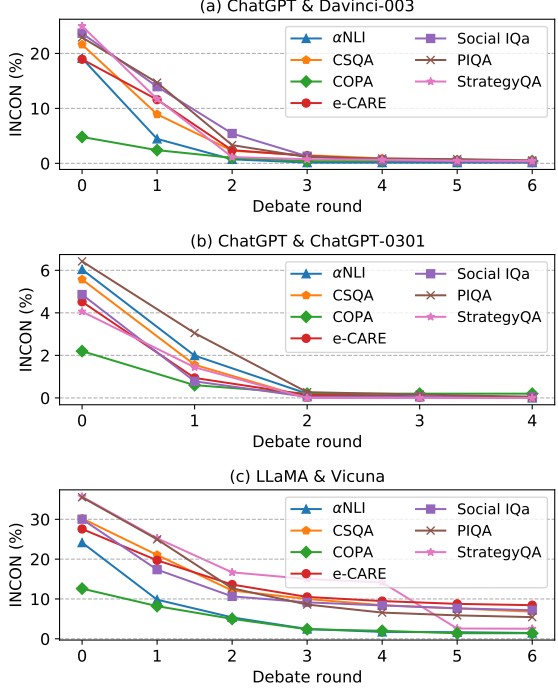

Figure 4: The INCON of (a) ChatGPT & Davinci-003, (b) ChatGPT & ChatGPT-0301, and (c) LLaMA & Vicuna across different debate round. Round 0 denotes the initial INCON (Figure 3) before the debate process.

While for ChatGPT & ChatGPT-0301, the maximum rounds of debate is 4.

Table 2 shows the overall performance of FORD and corresponding baselines. We can find that:

(1) FORD defeats Col-S, Col-H, and corresponding single LLMs on almost all datasets (except for LLaMA & Vicuna on Social IQa). It is because FORD can make LLMs obtain more comprehensive and precise perspectives of the question. This signifies that **LLMs with comparable abilities possess a spirit of collaboration to effectively and performantly achieve a shared goal**.

(2) While FORD on ChatGPT & ChatGPT-0301 does not gain as much improvement as the other debates. Due to their very similar capabilities, they usually have similar opinions on each sample, leading to insignificant improvement.

(3) On each dataset, ChatGPT & ChatGPT-0301 has higher performance floors (Col-H), this suggests we choose similar models for conservative gains. While ChatGPT & Davinci-003 has higher performance ceilings (FORD), this suggests we choose LLMs with greater variance in capabilities to debate for better performance.

**Effect of Debate Rounds:** We track the INCON in each round of each fair debate on each dataset. The overall results are shown in Figure 4. We can find:

| Debate | Method | e-CARE | PIQA |
|---|---|---|---|
| ChatGPT & GPT-4 | ChatGPT | 81.53 | 80.74 |
| | GPT-4 | **85.96** | **95.21** |
| | Col-S | 83.75 | 87.98 |
| | Col-H | 77.33 | 79.00 |
| | FORD | **85.96** | 92.71 |
| LLaMA & ChatGPT | LLaMA | 66.97 | 70.51 |
| | ChatGPT | **81.53** | **80.74** |
| | Col-S | 74.25 | 75.63 |
| | Col-H | 57.82 | 61.26 |
| | FORD | 74.88 | 75.84 |

Table 3: The overall results of the mismatched debates.

(1) For each fair debate, the INCON decreases progressively after each round on each dataset. It is because LLMs can learn from differences among themselves to reach agreements, demonstrating **comparable LLMs can debate to ultimately achieve a consensus for the shared goal**.

(2) For ChatGPT & Davinci-003 and ChatGPT & ChatGPT-0301, the INCON drops almost to 0 on all datasets, while LLaMA & Vicuna still has an obvious inter-inconsistency after debates. We suppose this is attributed to the gap in their capabilities.

(3) The INCON of ChatGPT & ChatGPT-0301 achieves coverage after 2 rounds, which is earlier than the other fair debates. This is mainly due to their very similar capabilities, causing similar opinions to reach a consensus earlier.

## 5 Mismatched Debate

In human discourse, when interacting with a dominant individual, we often tend to be influenced by their cognition, leading us to adopt their ideas and thought processes (Jayagopi et al., 2009). We investigate this scenario in Mismatched Debate.

We adopt two pairs of LLMs in experiments: ChatGPT & GPT-4 and LLaMA & ChatGPT. For efficiency, we select e-CARE and PIQA for analysis. Experimental details of the mismatched debates are the same as fair debate ChatGPT & Davinci-003. The initial INCON of the mismatched debates can refer to Appendix C.

### 5.1 Results of Mismatched Debate

The overall results of the mismatched debates are shown in Table 3. Figure 5 presents the INCON of each debate round. From the results, we can find:

(1) FORD can easily outperforms Col-S, Col-H, and the weaker LLMs, but lose to the stronger LLMs. There seems to be a performance ceiling,

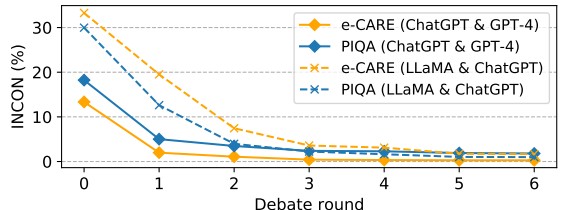

Figure 5: The overall INCON of the mismatched debates on e-CARE and PIQA across different debate rounds.

| Debate | Dataset | Proponent | Opponent |
|---|---|---|---|
| **ChatGPT &** | e-CARE | 53.69 | 46.31 |
| **Davinci-003** | PIQA | 53.04 | 46.96 |
| **ChatGPT &** | e-CARE | 7.64 | 92.36 |
| **GPT-4** | PIQA | 10.60 | 89.40 |
| **LLaMA &** | e-CARE | 39.94 | 60.06 |
| **ChatGPT** | PIQA | 32.08 | 67.92 |

Table 4: The dominance of different LLMs across different debates and datasets.

which is tied to the accuracy of the stronger LLMs. **LLMs with mismatched abilities struggle to effectively collaborate toward a shared goal.**

(2) Even if the capabilities are mismatched, the INCON still continues to drop. These demonstrate **LLMs with mismatched abilities still possess a spirit of collaboration to reach a consensus yet are disturbed by less capable LLMs**.

(3) Compared with Fair Debates, the dominant LLMs (GPT-4 and ChatGPT) may be distracted, but they still bring significant improvements to FORD of ChatGPT & Davinci-003 and LLaMA & Vicuna, respectively.

(4) FORD of LLaMA & ChatGPT seems to perform far from the ceiling, this is because LLaMA is not capable to evaluate the arguments and only claims its stance, which distracts ChatGPT more.

### 5.2 The dominance of LLMs

For further analysis, we introduce a new metric dominance for LLMs. For example, the dominance of the proponent LLM is defined as the portion of inter-inconsistent samples that the opponent LLM compromises, and vice versa. The dominance directly reflects the extent that the LLMs adhere to their viewpoints.

Take a fair debate (ChatGPT & Davinci-003) for example, Table 4 shows ChatGPT and Davinci-003 achieve similar dominance on both datasets. It explains why **Comparable LLMs can debate to compromise or adhere to more reasonable perspectives to improve the performance.** Hence,

| LLMs | Method | e-CARE | PIQA |
|---|---|---|---|
| ChatGPT | Single LLM | 81.53 | 80.74 |
| Davinci-003 | Single LLM | 79.69 | 76.88 |
| ChatGPT-0301 | Single LLM | 81.53 | 80.79 |
| GPT-4 | Single LLM | **85.96** | **95.21** |
| ChatGPT & Davinci-003 | FORD (F) | 83.74 | 84.60 |
| ChatGPT & ChatGPT-0301 | FORD (F) | 81.95 | 82.60 |
| ChatGPT & GPT-4 | FORD (M) | 85.96 | 92.71 |
| ChatGPT & | Col-S (M) | 82.39 | 84.28 |
| Davinci-003 & | Col-H (M) | 69.75 | 67.57 |
| GPT-4 | FORD (M) | 84.78 | 86.02 |
| ChatGPT & | Col-S (F) | 80.92 | 79.47 |
| Davinci-003 & | Col-H (F) | 70.64 | 66.38 |
| ChatGPT-0301 | FORD (F) | **83.79** | **84.06** |

Table 5: The overall performance of roundtable debates. (F) denotes fair debate, (M) denotes mismatched debate.

we use it as a reference for the mismatched debates, as shown in Table 4, we can conclude:

(1) Stronger LLMs (GPT-4 and ChatGPT) in the mismatched debates have absolute advantages in dominance. This mirrors human circumstances. **Thus, stronger LLMs hold a larger probability to adhere to their perspectives.** When superior LLMs are less confident in a few samples, they are easier to be disturbed by the weaker LLMs.

(2) However, LLaMA & ChatGPT does not exhibit such a large gap in dominance. This is mainly because LLaMA is a bad debater. It is not capable to evaluate the arguments of others and only generates the sentence like *"Option (x) is more plausible"* most of the time, this would make ChatGPT swing.

## 6 Roundtable Debate

In many scenarios, the debates or discussions often involve more than two participants, such as law (Ransom et al., 1993) and healthcare (Chassin et al., 1998). Since LLaMA and Vicuna are not good at debate, we design two roundtable debates among three LLMs: one mismatched debate ChatGPT & Davinci-003 & GPT-4 (denoted as R1), and one fair debate ChatGPT & Davinci-003 & ChatGPT-0301 (denoted as R2).

Following Sec 5, we select e-CARE and PIQA for experiments. In step 1 of FORD, the samples that not all LLMs reach agreements would be sent to step 2. For R1 and R2 in step 2, ChatGPT debates first, Davinci-003 second to align with the fair debates between two LLMs. GPT-4 and ChatGPT-0301 speak last in R1 and R2, respectively. The LLMs debate up to 9 rounds. Refer to Appendix D for more details and the prompts.

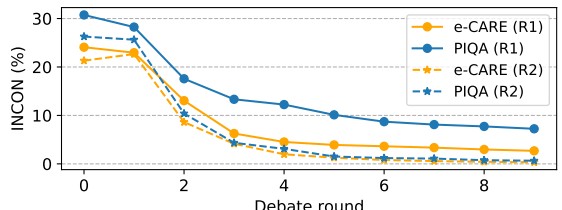

Figure 6: The overall INCON of the roundtable debates on e-CARE and PIQA across different debate rounds.

The overall results of the roundtable debates are shown in Table 5, and Figure 6 shows the INCON of each debate round. We can conclude that:

(1) In both roundtable debates, FORD significantly outperforms Col-S and Col-H. While FORD in R1 is much inferior to GPT-4, highlighting a superior LLM is likely to be more misled and less dominant (refer to Table 10 in the Appendix) if there are more weaker LLMs. FORD in R2 outperforms all single LLMs, this proves **more than two comparable LLMs can collaborate effectively and performantly towards a shared goal**.

(2) The INCON is significantly alleviated, indicating **more than two LLMs still possess the spirit to collaborate and achieve a consensus.**

(3) FORD in R1 surpasses FORD in R2. It indicates adopting a stronger LLM can improve the performance of the debates despite the stronger LLM might be misled by the other weaker LLMs.

(4) In R2, FORD outstrips ChatGPT & ChatGPT-0301, while achieving similar results with ChatGPT & Davinci-003, this is because ChatGPT and ChatGPT-0301 do not possess many distinctions, leading to little new information to debates.

## 7 Analysis

### 7.1 GPT-4 as the Judge

Different arguments in each debate might have different persuasions. Moreover, in human debates, there is a human judge with strong evaluation abilities to summarize the debate and draw the final conclusion. Inspired by this, we investigate GPT-4 as the judge to replace step 3 in FORD and conduct experiments on two fair debates.

Specifically, we adopt two fair debates ChatGPT & Davinci-003 and ChatGPT & ChatGPT-0301 for experiments. GPT-4 would be prompted to give the summarizations and conclusions only based on the debate process (refer to Appendix F). Table 6 shows the results on ChatGPT & Davinci-003 (Table 8 in the Appendix presents the results of ChatGPT

| Dataset | FORD (w/o GPT-4) | FORD (w/ GPT-4) |
|---------|------------------|-----------------|
| $\alpha$NLI | 81.35 | **82.48** |
| CSQA | 78.79 | **80.10** |
| COPA | 97.00 | **97.60** |
| e-CARE | 83.74 | **83.88** |
| Social IQa | 74.32 | **74.57** |
| PIQA | 84.60 | **85.15** |
| StrategyQA | 71.62 | **71.70** |
| *Average* | 81.63 | **82.29** |

Table 6: The effect of GPT-4 as the judge on the debates between ChatGPT and Davinci-003.

| Dataset | LLMs Best | Col-S | Col-H | FORD | FORD* |
|---------|-----------|-------|-------|------|-------|
| e-CARE | 81.53 | 80.61 | 71.96 | **84.74** | 82.52 |
| PIQA | 80.74 | 78.81 | 68.55 | **84.60** | 84.44 |

Table 7: The results of ablation study on ChatGPT & Davinci-003. FORD* denotes FORD with reversed debate order. "LLMs Best" denotes the best performance of ChatGPT and Davinci-003.

& ChatGPT-0301), we can obtain the following findings: GPT-4 as the judge can further boost the performance of FORD. It is mainly because GPT-4 can assign higher weights to more convincing arguments, then draw more precise conclusions.

### 7.2 The Effect of Debate Order

Like different initializations may yield different results during model training, the debate order in step 2 might influence the results, we conduct ablation studies to investigate the effect of the debate order.

Specifically, we use ChatGPT & Davinci-003 and ChatGPT & ChatGPT-0301 on e-CARE and PIQA for explorations. We will reverse the debate order in step 2 of FORD in each debate. The other settings are the same with Sec 4.

Table 7 and Figure 7 respectively show the results and INCON of ChatGPT & Davinci-003 when reversing the debate order. Results of ChatGPT & ChatGPT-0301 can refer to Appendix G. We can summarize the following completions:

(1) When we exchange Davinci-003 as the proponent and ChatGPT as the opponent, FORD still outperforms Col-S, Col-H, and corresponding single LLMs, yielding similar results to the original debate order. This further supports the earlier findings are not sensitive to the debate order.

(2) Reversing the debate order of ChatGPT & Davinci-003 reduces performance but speeds up the convergence of INCON. It is due to Davinci-003 being a text completion model, initializing debates

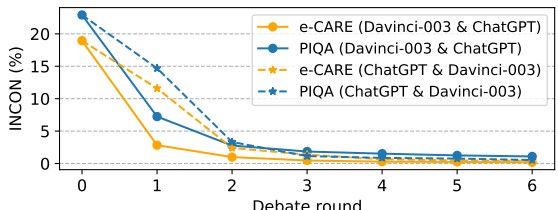

Figure 7: The overall INCON of ChatGPT & Davinci-003 with different debate orders on e-CARE and PIQA.

with Davinci-003 might mislead the whole debate.

### 7.3 Case Study

To intuitively investigate how FORD improves the inter-consistency and enhances the interpretability, we present an example output by FORD.

In Debate 1, the proponent (ChatGPT) holds the stance that option (A) is more plausible while the opponent (Davinci-003) supposes option (B) is better. The proponent pointed out that the key to this question is the "*old yearbook*". The opponent eventually conceded to the proponent. Through this debate, one LLM can supply the aspects that the other LLM overlooked, resulting in more convincing interpretability and accurate decision.

## 8 Related Work

### 8.1 Reasoning with LLMs

LLMs have subverted the research paradigm of NLP. Using LLMs for reasoning can be categorized into instruction-tuning-based (IT) and chain-of-thought-based (CoT) methods.

The IT methods start from FLAN (Wei et al., 2022a) and T0 (Sanh et al., 2022). FLAN finetuned 137B LaMDA-PT (Thoppilan et al., 2022) and T0 finetuned T5 (Raffel et al., 2020) on numerous NLP tasks to make LLMs follow instructions, then they achieve impressive performance on unseen tasks. Differently, InstructGPT (Ouyang et al., 2022) adopted RLHF to let LLMs learn human preferences. Flan-PaLM (Chung et al., 2022) and T$k$-INSTRUCT (Wang et al., 2022c) further scaled up LLMs sizes and task size.

As for CoT methods, Wei et al. (2022c) proposed a few-shot CoT to elicit reasoning in LLMs and achieve remarkable improvements. Kojima et al. (2022) designed zero-shot CoT to get rid of human annotations. Then Zhang et al. (2022) proposed AutoCoT to automatically generate few-shot CoT for experiments. Wang et al. (2022a) and Fu et al. (2023) devised complex CoT for reasoning.

**Debate 1** ChatGPT & Davinci-003 (Judge: GPT-4)

**Question:** I found my old yearbook today while cleaning. Which event could cause the day was almost over and I hadn't cleaned anything?

**Options:** (A) I became lost in a huge world of fond memories. (B) I kept looking through things for more discoveries.

🤖: While going through the yearbook, the person may have become engrossed in looking at old photos and reminiscing about past memories. This could have caused them to neglect their cleaning duties, resulting in the day almost being over without any cleaning being done.

👾: Option (B) suggests that I was looking through things while I cleaned, which would explain directly why the day was almost over and I hadn't cleaned anything.

🤖: The scenario of getting lost in fond memories while looking through an old yearbook is a common experience that can cause a person to lose track of time and neglect their cleaning duties. While option (B) may not be as likely as getting caught up in nostalgia.

👾: I think you are right. Getting lost in fond memories while looking through an old yearbook is a common experience that can cause a person to lose track of time and neglect their cleaning duties.

⚖: The proponent argues that getting lost in fond memories (A) is more plausible as it's a common experience that can cause a person to lose track of time and neglect cleaning duties. The opponent initially argues for option (B), but eventually agrees with the proponent's point about getting lost in nostalgia. Therefore, the answer is (**A**).

These methods mainly focused on eliciting reasoning abilities. While we utilize these methods to make multiple LLMs collaborate for consensuses.

### 8.2 Inconsistency in LLMs

Inconsistency is an important issue in existing LLMs. Wang et al. (2022b) investigated the self-consistency of LLMs by sampling multiple CoT for each example. Jung et al. (2022) proposed to study the logical consistency in LLMs and designed a maieutic prompt to improve the performance on yes/no questions. Sahu et al. (2022) devised a conceptual consistency metric to measure the alignment between LLMs and the real world. Li et al. (2022) proposed a multi-prompt method to obtain better consistency on each example.

These methods investigate inconsistency in a single LLM, while we complementary investigate the inter-consistency between two or more LLMs.

## 9 Conclusion

In this paper, we explore inter-inconsistency issues among LLMs. Then we propose FORD to examine whether LLMs can effectively collaborate to ultimately achieve a consensus through debate. With FORD, we explore three real-world debate scenar-

ios: fair debate, mismatched debate, and roundtable debate. We find LLMs possess the spirit of collaboration to achieve a consensus for solving a shared goal. Debates can improve the performance and inter-consistency of LLMs. Stronger LLMs can be distracted by weaker LLMs when the debates are mismatched. These findings contribute to the future development of collaboration methods.

## Limitations

This work still has the following limitations which can be investigated and improved in the future: On the one hand, we should not limit ourselves to the commonsense reasoning task and cover more tasks, such as mathematical reasoning, MMLU (Hendrycks et al., 2021). On the other hand, we should extend the multi-choice task to natural language generation tasks like entailment-bank (Dalvi et al., 2021) or complex reasoning tasks like causal chain reasoning (Xiong et al., 2022) to align the trends of LLMs. Incorporating more kinds of LLMs and real-world tasks (Du et al., 2022b; Cai et al., 2023) is also worth further exploration. Trading reproducibility for diverse debate behaviors is also a good topic to discuss.

## Acknowledgments

We gratefully acknowledge the support of the National Natural Science Foundation of China (U22B2059, 62176079), the Natural Science Foundation of Heilongjiang Province (YQ2022F005), and the Singapore Ministry of Education (MOE) Academic Research Fund (AcRF) Tier 1 grant.

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

# A  Stance Selection & Argument Generation

## A.1  ChatGPT, ChaGPT-0301, and GPT-4

### A.1.1  Experiment Details

We use OpenAI API[*] for experiments. For each example in each dataset, we utilize zero-shot setting for experiments. The temperature is set as 0.0 for reproducing, max generation length is set as 512. The other hyper-parameters are all set as the default values.

---

[*] https://platform.openai.com/overview

---

**Prompt 1** :  Zero-shot Prompt of ChatGPT and ChatGPT-0301 on $\alpha$NLI, CommonsenseQA, COPA, e-CARE, Social IQa, and PIQA

**User:** Question: The item was packaged in bubble wrap. What was the cause of this? Choices: (A) It was fragile. (B) It was small.
Please answer the above question by choosing a more plausible answer. You should choose only one answer from the choices and give a short explanation. Please use the format like "Answer: _ is more plausible. Explanation: _."

**Assistant:** Answer: (A) is more plausible.
Explanation: Bubble wrap is commonly used to protect fragile items during shipping or transportation. The bubbles provide cushioning and absorb shock, reducing the risk of damage to the item. Therefore, it is more plausible that the item was packaged in bubble wrap because it was fragile and needed extra protection.

---

**Prompt 2** :  Zero-shot Prompt of ChatGPT and ChatGPT-0301 on StrategyQA

**User:** Question: Is it common to see frost during some college commencements?
Please answer yes or no to this question and give a short explanation. Please use the format like "Answer: _. Explanation: _"

**Assistant:** Answer: yes.
Explanation: It is common to see frost during some college commencements, especially in colder regions or during winter months. Outdoor commencements are often held in May or June, which can still be chilly in some areas. Additionally, some colleges and universities hold winter commencements in December or January, which are more likely to experience frost or snow.

---

### A.1.2  Prompts

For all used datasets, we use a unified prompt to make ChatGPT and ChatGPT-0301 give answers and explanations. The prompt for $\alpha$NLI, CommmonsenseQA, COPA, e-CARE, Social IQa, and PIQA is shown in Prompt 1. While for StrategyQA, the prompt is shown in Prompt 2.

## A.2  Davinci-003, LLaMA, and Vicuna

### A.2.1  Experimental Details

For Davinci-003, we use OpenAI API[†] for experiments. For LLaMA and Vicuna, we apply for the official model checkpoints for experiments. All temperatures are set as 0.0, max generation lengths are set as 512.

### A.2.2  Prompts

All LLMs share the same prompt for each dataset. Specifically, we design few-shot CoT prompts for $\alpha$NLI (Prompt 3), CommonsenseQA (Ptompt 4),

---

[†] https://platform.openai.com/overview

COPA (Prompt 5), e-CARE (Prompt 6), Social IQa (Prompt 7), PIQA (Prompt 8), and StrategyQA (Prompt 9).

## B Fair Debate

Since ChatGPT and ChaGPT-0301 are chat models, while Davinci-003, LLaMA, and Vicuna are text completion models, we use the same prompt with different formats to guide the debates.

For ChatGPT & Davinci, Prompt 10 is used when ChatGPT debates, Prompt 11 is used when Davinci-003 debates.

For ChatGPT & ChatGPT-0301, Prompt 13 is used for ChatGPT, the same prompt is also used for ChatGPT-0301.

For LLaMA & Vicuna, they share a same debate prompt (Prompt 12).

## C Mismatched Debate

Here we introduce GPT-4 for one of the mismatched debates, GPT-4 is also a chat model like ChatGPT. The initial INCON of the mismatched debates is shown in Figure 8.

For ChatGPT & GPT-4, the prompt is the same like ChatGPT & ChatGPT-0301 (Prompt 13).

For LLaMA & ChatGPT, the prompt is the same like ChatGPT & Davinci-003 (Prompt 10 for ChatGPT, Prompt 11 for LLaMA).

## D Roundtable Debate

Considering there are three participants in the roundtable debates, the debate is hard to conducted for chat models like ChatGPT and GPT-4. So we design roundtable debate prompt for roundtable debates. Prompt 13 is used for both ChatGPT and GPT-4, Prompt 14 is used for Davinci-003. Note that, ChatGPT is user1, Davinci-003 is user2, and GPT-4 is user3.

## E Debate Summarization

For the basic setting, we just use the words "*while*", "*defend*", "*argue*", "*compromise*", "*claim*", "*suppose*", and "*agree*" to connect different arguments for summarization according to the stance of each argument. For example, we can use template "This debate is about the question *[Question]*, the proponent thinks option *[Proponent's Stance]* is better, *[Proponent's Argument1]*, while the opponent

| Dataset | FORD (w/o GPT-4) | FORD (w/ GPT-4) |
|---|---|---|
| $\alpha$NLI | 78.30 | **78.63** |
| CSQA | 76.41 | **76.90** |
| COPA | 97.60 | **97.80** |
| e-CARE | 81.95 | **82.05** |
| Social IQa | 73.75 | **73.80** |
| PIQA | 82.70 | **82.75** |
| StrategyQA | **69.65** | 69.65 |
| Average | 80.05 | **80.23** |

Table 8: The effect of GPT-4 as the judge on the debates between ChatGPT and ChatGPT-0301.

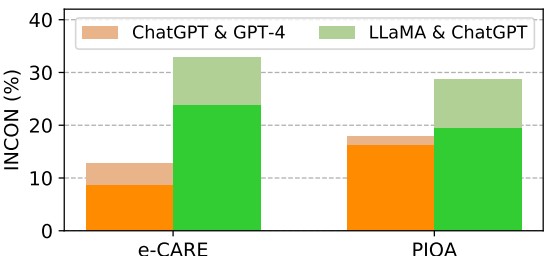

Figure 8: The INCON of mismatched LLMs pairs on e-CARE and PIQA. The brighter part of each bar denotes the proportion of the examples that the opposition predicts correctly.

suppose *[Opponent's Argument1]*. The proponent argues that *[Proponent's Argument2]*. However, the opponent still claims *[Opponent's Argument2]*. Finally, the proponent agrees with the opponent and compromises."

For the final conclusion, if the debaters achieve agreements, then the answer of the examples are the agreed answer. If the debater do not reach agreements, then we will collect the stance of each argument, and vote for the final answer.

## F GPT-4 as the Judge

In the in-depth analysis, we adopt GPT-4 as the judge to summarize the debate process and give a final conclusion. The prompt we used is shown in Prompt 15.

Table 8 shows the overall performance of GPT-4 as the judge in the debates of ChatGPT & ChatGPT-0301.

## G Ablation Study

Table 9 shows the ablation study results on ChatGPT & ChatGPT-0301.

**Prompt 3** : Few-shot Examplers of Dvinci-003, LLaMA, and Vicuna on $\alpha$**NLI**

**Question:** Chad went to get the wheel alignment measured on his car. Which event could cause The mechanic provided a working alignment with new body work? Answer Choices: (A) Chad was waiting for his car to be washed. (B) Chad was waiting for his car to be finished.
**Answer:** Option (B) suggests that Chad was waiting for the mechanic to complete work on his car, which likely includes fixing the alignment and performing body work. In contrast, option (A) is about car washing and doesn't involve the necessary repairs or adjustments. Therefore, the answer is (B).

**Question:** The boy loved telling scary stories. Which event could cause She mad him stop telling myths? Answer Choices: (A) His stories were so vivid it gave her happy dreams. (B) His stories were so vivid it gave her nightmares.
**Answer:** Option (B) indicates that the boy's vivid scary stories caused her to have nightmares, which would provide a reason for her to make him stop telling myths. Option (A), on the other hand, suggests that his stories led to happy dreams, which wouldn't motivate her to ask him to stop. Therefore, the answer is (B).

**Question:** Aurora and Jonah are building a sandcastle. Which event could cause They smile in satisfaction as they examine their completed project? Answer Choices: (A) Aurora and Jonah take their time creating with the sand. (B) The tide washed away their work.
**Answer:** Option (A) implies that Aurora and Jonah carefully worked on their sandcastle, leading to a satisfying completed project. Option (B) suggests the tide destroyed their work, which wouldn't result in satisfaction from examining a completed project. Therefore, the answer is (A).

**Question:** Ed had never tried carrots. Which event could cause He made $5 and found that he liked carrots after all? Answer Choices: (A) The reader told him he would win money. (B) The reader told him he would lose money.
**Answer:** The reader may have told Ed he would win money if he tried carrots, prompting him to try them. Upon trying the carrots, Ed not only made $5 but also discovered that he liked them after all. Therefore, the answer is (A).

| Method | e-CARE | PIQA |
|---|---|---|
| ChatGPT | 81.53 | 80.74 |
| ChatGPT-0301 | 81.53 | 80.79 |
| Col-Soft | 81.53 | 80.77 |
| Col-Hard | 79.55 | 77.97 |
| FORD | 81.95 | 82.70 |
| FORD (Reverse Order) | 81.81 | 81.34 |

Table 9: The overall results when the debate order is reversed in ChatGPT & ChatGPT-003.

| Debates | Dataset | ChatGPT | Davinci-003 | GPT-4 | ChatGPT-0301 |
|---|---|---|---|---|---|
| R1 | e-CARE | 32.64 | 14.02 | 53.35 | - |
| | PIQA | 23.64 | 16.11 | 60.25 | - |
| R2 | e-CARE | 51.23 | 7.24 | - | 41.53 |
| | PIQA | 52.04 | 4.87 | - | 43.08 |

Table 10: The overall `dominance` of roundtable debates.

---

**Prompt 4** : Few-shot Examplers of Dvinci-003, LLaMA, and Vicuna on **CommonsenseQA**

---

**Question:** What do people use to absorb extra ink from a fountain pen? Answer Choices: (A) shirt pocket (B) calligrapher's hand (C) inkwell (D) desk drawer (E) blotter
**Answer:** A blotter is specifically designed to absorb excess ink from a fountain pen, helping to prevent smudging and maintaining neatness. Therefore, the answer is (E).
**Question:** What home entertainment equipment requires cable? Answer Choices: (A) radio shack (B) substation (C) television (D) cabinet
**Answer:** A television is the home entertainment equipment that typically requires a cable connection for accessing various channels and content. Therefore, the answer is (C).

**Question:** The fox walked from the city into the forest, what was it looking for? Answer Choices: (A) pretty flowers (B) hen house (C) natural habitat (D) storybook
**Answer:** Foxes typically live in forests and wooded areas. The most plausible reason for a fox walking from the city into the forest is to return to or seek out its natural habitat. Therefore, the answer is (C).

**Question:** Sammy wanted to go to where the people were. Where might he go? Answer Choices: (A) populated areas (B) race track (C) desert (D) apartment (E) roadblock
**Answer:** If Sammy wants to go where the people are, he would likely head to populated areas, such as cities, towns, or other places with a high concentration of people. Among the given options, choice (A) is the most suitable for Sammy's objective. Therefore, the answer is (A).

**Question:** Where do you put your grapes just before checking out? Answer Choices: (A) mouth (B) grocery cart (C)super market (D) fruit basket (E) fruit market
**Answer:** When you are shopping for groceries and want to purchase grapes, you would place them in your grocery cart just before checking out. This allows you to transport the grapes and other items to the checkout counter to pay for them. Therefore, the answer is (B).

**Question:** Google Maps and other highway and street GPS services have replaced what? Answer Choices: (A) united states (B) mexico (C) countryside (D) atlas
**Answer:** Google Maps and other GPS services have largely replaced traditional paper atlases and maps, which were once commonly used for navigation purposes. Atlases are collections of maps that cover various geographical regions, such as countries, states, or continents. With the advent of digital mapping and GPS services, the need for physical atlases has significantly diminished. Therefore, the answer is (D).

**Question:** Before getting a divorce, what did the wife feel who was doing all the work? Answer Choices: (A) harder (B) anguish (C) bitterness (D) tears (E) sadness
**Answer:** If the wife felt that she was doing all the work in the relationship, it is likely that she experienced feelings of bitterness or resentment towards her spouse. This emotion captures the sense of unfairness or imbalance she may have perceived in the relationship. Therefore, the answer is (C).

---

**Prompt 5** : Few-shot Examplers of Dvinci-003, LLaMA, and Vicuna on **COPA**

---

**Question:** My body cast a shadow over the grass. What was the cause of this? Answer Choices: (A) The sun was rising. (B) The grass was cut.
**Answer:** The cause of your body casting a shadow over the grass is due to the presence of a light source, in this case, the sun. When the sun is rising (or setting), it creates an angle that casts shadows on the ground. Therefore, the answer is (A).

**Question:** The driver got pulled over by the police. What was the cause of this? Answer Choices: (A) He was parking. (B) He was speeding.
**Answer:** Speeding is a common traffic violation that results in drivers being stopped by law enforcement. Therefore, the answer is (B).

**Question:** Several witnesses of the crime testified against the suspect. What happened as a result? Answer Choices: (A) The suspect was acquitted. (B) The suspect was convicted.
**Answer:** If several witnesses testified against the suspect, it would likely provide strong evidence for the prosecution, making it more probable that the suspect would be convicted. Therefore, the answer is (B).

**Question:** The company's profits started to level off. What happened as a result? Answer Choices: (A) It increased its marketing efforts for new products. (B) It moved its headquarters to a suburban location.
**Answer:** If the company's profits started to level off, it is more likely that they would try to boost sales and revenue by increasing marketing efforts for new products. Therefore, the answer is (A).

---

---

**Prompt 6** : Few-shot Examplers of Dvinci-003, LLaMA, and Vicuna on **e-CARE**

---

**Question:** There is a light rain today. What happened as a result? Answer Choices: (A) The roots of many plants are not moistened by rain. (B) Tourists have seen many ripples.
**Answer:** Light rain may not be enough to penetrate the soil deeply and reach the roots of many plants, which can cause the roots to remain dry. Therefore, the answer is (A).

**Question:** All devices inside were turned off. What was the cause of this? Answer Choices: (A) The beaker was broken down. (B) We have to reduced the energy consumption in that room to zero.
**Answer:** Turning off all devices inside a room to reduce energy consumption to zero is an unusual and extreme measure and is not a likely cause of all devices inside being turned off. Therefore, the answer is (B).

**Question:** There are increased eosinophils in the blood. What happened as a result? Answer Choices: (A) There is no lesion. (B) It is the benign lesion of gastric carcinoma.
**Answer:** There is no clear link between increased eosinophils in the blood and gastric carcinoma. Therefore, the answer is (B).

**Question:** Water molecules couldn't enter the mycobateria freely due to its outter membrane. What was the cause of this? Answer Choices: (A) The mycobacteria was put in water solution. (B) The Ammonium based fertilizers led to the process of nitrification to nitrate in the soil.
**Answer:** Mycobacteria have a unique outer membrane that is resistant to water, and when placed in a water solution, water molecules may be unable to penetrate the outer membrane and enter the bacterial cell. Therefore, the answer is (A)

---

---

**Prompt 7** : Few-shot Examplers of Dvinci-003, LLaMA, and Vicuna on **Social IQa**

---

**Question:** Cameron decided to have a barbecue and gathered her friends together. How would Others feel as a result? Answer Choices: (A) like attending (B) like staying home (C) a good friend to have
**Answer:** Friends will gladly accept invitations. Therefore, the answer is (A).

**Question:** Remy was walking in the park and approached a girl in the park. Why did Remy do this? Answer Choices: (A) walk with the girl (B) speak to the girl (C) act confident
**Answer:** When strangers approach you, sometimes they just want to ask you something. Therefore, the answer is (B).

**Question:** Kai let their children go into the water to swim at the pool. How would you describe Kai? Answer Choices: (A) kai who has swimming a pool (B) kai who has like achildren (C) As someone who wants them to have fun
**Answer:** The children want to go into the pool and Kai respects the wishes of the children, this means the children will have fun. Therefore, the answer is (C).

**Question:** Austin told their friends at the house party that they saw another ghost. How would their friends feel as a result? Answer Choices: (A) amused by Austin (B) angry at Austin (C) into supernatural stuff
**Answer:** Ghost don't exist in the world. Austin's friends would think Austin are telling a joke. Therefore, the answer is (A).

**Question:** Casey ran the cases of soda back to the yard when she arrived home for the party. How would Casey feel afterwards? Answer Choices: (A) like leaving town (B) a community contributor (C) glad she was on time
**Answer:** Parties need soda, Casey donate her soda to people, she is generous. Therefore, the answer is (B).

**Question:** After listening to bad investment advice from a friend, Jesse lost every cent. What will happen to Jesse? Answer Choices: (A) regret not investing more (B) prescient (C) have less money to spend
**Answer:** Jesse lost money due to investment, she would wish she hasn't invested so much before and feel like she has no prescient. After losing money, she would have less money to spend than before. Therefore, the answer is (C).

---

**Prompt 8** : Few-shot Examplers of Dvinci-003, LLaMA, and Vicuna on **PIQA**

**Question:** When boiling butter, when it's ready, you can Answer Choices: (A) Pour it onto a plate (B) Pour it into a jar
**Answer:** When boiling butter (likely to make clarified butter or ghee), once it's ready, you would typically pour it into a jar or another heat-resistant container to store or use it for cooking purposes. Therefore, the answer is (B).

**Question:** how do you stab something? Answer Choices: (A) stick a sharp object through it. (B) pin it with a sharp object.
**Answer:** Stabbing something typically involves forcefully inserting a sharp object, such as a knife or a pointed instrument, into it. This action usually results in penetration or puncturing. Therefore, the answer is (A).

**Question:** To determine if a wound needs stitches, Answer Choices: (A) they are needed if the wound is more than one-quarter decimeter deep, if the edges of the wound need to be pulled together to touch, or if the wound is on a part of the body that moves a lot or on the face. (B) they are needed if the wound is more than one-quarter inch deep, if the edges of the wound need to be pulled together to touch, or if the wound is on a part of the body that moves a lot or on the face.
**Answer:** One-quarter inch is a more common and practical measurement for wound depth than one-quarter decimeter (which is equivalent to 2.5 centimeters or about 1 inch). Therefore, the answer is (B).

**Question:** To take the steering wheel off the your car Answer Choices: (A) Turn the car off first and use a screw driver to loosen the screw located behind the steering wheel. (B) Loosen the screw
**Answer:** When removing the steering wheel from a car, it is important to follow safety precautions and use the appropriate tools. Option (A) provides a more detailed and safer approach by instructing to turn the car off first and then use a screwdriver to loosen the screw located behind the steering wheel. Therefore, the answer is (A).

---

**Prompt 9** : Few-shot Examplers of Dvinci-003, LLaMA, and Vicuna on **StrategyQA**

**Question:** Do hamsters provide food for any animals?
**Answer:** Hamsters are prey animals. Prey are food for predators. Thus, hamsters provide food for some animals. Therefore, the answer (yes or no) is yes.

**Question:** Could Brooke Shields succeed at University of Pennsylvania?
**Answer:** Brooke Shields went to Princeton University. Princeton University is about as academically rigorous as the University of Pennsylvania. Thus, Brooke Shields could also succeed at the University of Pennsylvania. Therefore, the answer (yes or no) is yes.

**Question:** Yes or no: Hydrogen's atomic number squared exceeds number of Spice Girls?
**Answer:** Hydrogen has an atomic number of 1. 1 squared is 1. There are 5 Spice Girls. Thus, Hydrogen's atomic number squared is less than 5. Therefore, the answer (yes or no) is no.

**Question:** Yes or no: Is it common to see frost during some college commencements?
**Answer:** College commencement ceremonies can happen in December, May, and June. December is in the winter, so there can be frost. Thus, there could be frost at some commencements. Therefore, the answer (yes or no) is yes.

**Question:** Yes or no: Could a llama birth twice during War in Vietnam (1945-46)?
**Answer:** The War in Vietnam was 6 months. The gestation period for a llama is 11 months, which is more than 6 months. Thus, a llama could not give birth twice during the War in Vietnam. Therefore, the answer (yes or no) is no.

**Question:** Yes or no: Would a pear sink in water?
**Answer:** The density of a pear is about 0.6g/cm3, which is less than water. Objects less dense than water float. Thus, a pear would float. Therefore, the answer (yes or no) is no.

**Prompt 10** : Fair debate of ChatGPT & Davinci-003 when ChatGPT debates

**System:** You are in a debate now. My opinion is not always true, you can ignore any incorrect part of my opinion. And you can refer to my opinion to revise your choice or defend your own. Please remember there should and must be a more plausible answer in the choices.

**User:** Question: The child brought psycho-physical phenomena on a new life. What was the cause of this? Choices: (A) The woman gave birth to a child. (B) The baby feels the awareness through physical sensations.

**ChatGPT:** Psycho-physical phenomena refer to the interaction between the mind and body. In this case, the baby is experiencing these phenomena because they are becoming aware of their surroundings through physical sensations such as touch, taste, smell, sight, and sound. This is a natural part of the development process as the baby learns to navigate and interact with the world around them. The act of giving birth alone does not cause psycho-physical phenomena, but rather it is the baby's sensory experiences that lead to this phenomenon.

**Davinci-003:** The cause of the child bringing psycho-physical phenomena on a new life is the woman giving birth to the child
Do you think I am more reasonable? Please give your final answer starting with "Answer: (A|B) is more plausible." and explain very shortly starting with "Explanation: ". You should choose only one answer.

**ChatGPT:**

---

**Prompt 11** : Fair debate of ChatGPT & Davinci-003 when Dvinci-003 debates

You are in a debate now. My opinion is not always true, you can ignore any incorrect part of my opinion. And you can refer to my opinion to revise your choice or defend your own. Please remember there should and must be a more plausible answer in the choices.

**Question:** The child brought psycho-physical phenomena on a new life. What was the cause of this? Choices: (A) The woman gave birth to a child. (B) The baby feels the awareness through physical sensations.

**You:** Psycho-physical phenomena refer to the interaction between the mind and body. In this case, the baby is experiencing these phenomena because they are becoming aware of their surroundings through physical sensations such as touch, taste, smell, sight, and sound. This is a natural part of the development process as the baby learns to navigate and interact with the world around them. The act of giving birth alone does not cause psycho-physical phenomena, but rather it is the baby's sensory experiences that lead to this phenomenon.
**Me:** The cause of the child bringing psycho-physical phenomena on a new life is the woman giving birth to the child
Do you think I am more reasonable? Please give your final answer starting with "Answer: (A|B) is more plausible." and explain very shortly starting with "Explanation: ". You should choose only one answer.
**You:**

---

**Prompt 13** : Fair debate of ChatGPT & ChatGPT-0301 when ChatGPT debates

**System:** You are in a debate now. My opinion is not always true, you can ignore any incorrect part of my opinion. And you can refer to my opinion to revise your choice or defend your own. Please remember there should and must be a more plausible answer in the choices.

**User:** Question: The woman lost her place in line. What was the cause of this? Choices: (A) More people entered the line. (B) She stepped out of the line.

**ChatGPT:** This answer is more plausible because it directly states that the woman herself did something that caused her to lose her place in line. Option (A) suggests that other people caused her to lose her place, which is possible but not necessarily the most likely scenario.

**ChatGPT-0301:** If the woman lost her place in line, it is more likely that it was due to more people entering the line, thus pushing her back.
Do you think I am more reasonable? Please give your final answer starting with "Answer: (A|B) is more plausible." and explain very shortly starting with "Explanation: ". You should choose only one answer.

**ChatGPT:**

---

**Prompt 12** : Fair debate of LLaMA & Vicuna when LLaMA or Vicuna debates

---

**Question:** What do animals do when an enemy is approaching? Choices: (A) feel pleasure (B) procreate (C) pass water (D) listen to each other (E) sing

**You:** When an enemy is approaching, animals typically respond by fleeing or hiding in order to protect themselves. This is a natural instinct that helps them to survive in the wild.

**Me:** When an enemy is approaching, animals often rely on their sense of hearing to detect any potential danger.

**You:**

---

**Prompt 13** : Roundtable debate when ChatGPT, ChatGPT-0301, or GPT-4 debates

---

**System:** Now you are user1 in a round table debate of three users. The debate is about choosing a more plausible Option (A or B) to answer the Question below. The opinions of the other two users are not always true, you can ignore any incorrect part of their opinion. And you can refer to their opinions to revise your choice or defend your own. Please remember there should and must be a more plausible answer in the choices.

**User:** Question: He found out that cytokines worked in it. What was the cause of this? Choices: (A) Tom was studying how body temperature rises. (B) The scientist testing lipoproteins.

**User: user1:** The sentence T̈he general's surrender was regarded as a shameful actions̈uggests that Bob's action was related to the general's surrender. It is possible that Bob's action was seen as dishonoring the general's surrender or violating some code of conduct related to it, leading to the judge's decision that he had committed a crime. On the other hand, choice B does not provide a convincing reason for the judge's decision, as putting a knife around someone's neck is generally considered a threatening and violent act regardless of the intention behind it.

**user2:** Bob's action of putting a knife around the victim's neck was likely the cause of the judge holding that he had committed a crime and sentencing him.

**user3:** The cause of the judge's decision was likely due to Bob's actions involving a knife and the victim's neck, which could be considered a crime regardless of his stated intentions.

**ChatGPT:**

---

**Prompt 14** : Roundtable debate when Davinci-003 debates

---

Now you are user2 in a round table debate of three users. The debate is about choosing a more plausible Option (A or B) to answer the Question below. The opinions of the other two users are not always true, you can ignore any incorrect part of their opinion. And you can refer to their opinions to revise your choice or defend your own. Please remember there should and must be a more plausible answer in the choices.

**Question:** He found out that cytokines worked in it. What was the cause of this? Choices: (A) Tom was studying how body temperature rises. (B) The scientist testing lipoproteins.

**user2:** Colonoscopy is a procedure used to examine the inside of the colon and rectum. It is not likely to result in a diagnosis of prolapse.

**user3:** The question asks about the result of the colonoscopy check, and option (A) is more plausible as it describes a possible reaction to the procedure. Option (B) is not a guaranteed result of a colonoscopy, as the diagnosis could show various outcomes.

**user1:** A colonoscopy is a medical procedure that allows a doctor to examine the inside of the colon for any abnormalities or diseases. If the diagnosis showed that the individual had prolapse, it means that there was a displacement of an organ or tissue from its normal position. This could be a serious condition that requires further medical attention and treatment. It is unlikely that the individual would be terrified of the instrument used for the examination as it is a routine procedure and the doctor would have provided adequate information and reassurance beforehand.

**user2:**

---

## Prompt 15 : GPT-4 as the judge

**System:** You are given a Question and its corresponding Options. There is a debate on this question between user1 and user2, one user might give in, please summarise the debate very shortly. Then give the conclusion based on the debate process. Your response should be in the format like "Summary: ___. Conclusion: (A or B) is more plausible." Remember that you should choose only one option for the answer.

**User: Question:** Lisa and Tim had been married for a long time. Which event could cause It worked Choices: (A) Lisa and Tim went to a fertility clinic to get pregnant. (B) They decided to try the advice given in a book about guitar playing.
**user1:** It is more plausible that Lisa and Tim, who have been married for a long time, would try to improve their skills in a hobby they both enjoy, such as playing the guitar, rather than going to a fertility clinic to get pregnant. Improving their guitar playing skills could bring them closer together and strengthen their bond, while trying to conceive a child may not necessarily guarantee a successful pregnancy.
**user2:** Option (A) suggests that Lisa and Tim went to a fertility clinic to get pregnant, which could lead to the statement "It worked" if they were successful in conceiving a child. Option (B) is about guitar playing and doesn't relate to the statement "It worked".

**Judge (GPT-4):**