# OpenReview forum: "Examining Inter-Consistency of Large Language Models Collaboration: An In-depth Analysis via Debate"
_EMNLP/2023/Conference — EMNLP 2023 Findings_

### Official Review · Reviewer_deKp · 2023-08-04

**Soundness:** 4

**Excitement:**

4: Strong: This paper deepens the understanding of some phenomenon or lowers the barriers to an existing research direction.

**Paper Topic And Main Contributions:**

This paper introduces a new framework FORD to aid LLMs in their reasoning process. Specifically, in FORD, LLMs are allowed to explore the differences between their understandings and the conceptualizations of others via debate. This new method can provide performance gains through mutual learning. At last, FORD’s efficacy and LLMs’ property are empirically proved and discovered through comprehensive experiments on seven datasets.

**Questions For The Authors:**

1. Could the FORD method outperform existing single-LLM reasoning methods?
2. How is the cost of FORD compared with existing single-LLM reasoning methods?

**Reasons To Accept:**

1. The related works that the paper cites are up-to-date, thorough, and high-quality from top conferences. And the logic, diagrams, and notations are clear and easy to read.
2. This paper introduces the structure of debate for LLMs collaboration. Its universality and superiority in seven commonsense reasoning datasets show its significance.
3. The paper evaluates the method on the dataset and achieves better results than baselines. The experiments in the paper are extensive and convincing. The chosen evaluation metrics are presentative, and there are ablation experiments showing the significance of using the debate structure.
4. This work discovers that LLMs can collaborate with other LLMs to performantly achieve a shared goal.

**Reasons To Reject:**

1. The main concern is that the experimental comparison is inadequate. As mentioned in the motivation, existing studies focus on the inconsistency issues within a single LLM. It is unclear whether this kind of multi-agent detate is better. Therefore, it is still worth comparing the proposed method with these kinds of existing methods. For example, Self-consistency or Self-Refine ([https://arxiv.org/abs/2303.17651](https://arxiv.org/abs/2303.17651)), in which the LLM provides arguments and explanations to itself.
2. Such approach may be much more costly than single-LLM methods.


**Reproducibility:**

4: Could mostly reproduce the results, but there may be some variation because of sample variance or minor variations in their interpretation of the protocol or method.

**Reviewer Confidence:**

4: Quite sure. I tried to check the important points carefully. It's unlikely, though conceivable, that I missed something that should affect my ratings.

---

> ### Author Rebuttal · Authors · 2023-08-29
>
> Thanks for the valuable feedback and insightful comments!
> ## ● Response 1 (Reasons to Reject 1)
> | Method                         | $\alpha$NLI  | e-CARE |
> | ------------------------------ | ----- | ------ |
> | Self-Consistency (ChatGPT)     | 77.84 | 82.47  |
> | Self-Consistency (Davinci-003) | 79.89 | 76.44  |
> | FORD (ChatGPT & Davinci-003, Ours)   | **81.35** | **83.74**  |
>
> ***Table 1: The comparisons between self-consistency and FORD***
>
> Thanks for the suggestions, we can add the baselines of self-consistency or self-refine to better understand the effects of LLMs collaborations. We first randomly choose two datasets and one fair debate (ChatGPT & Davinci-003) for the comparison with self-consistency (sample 3 arguments for a fair comparison with FORD). Table 1 above shows the whole results, FORD show advantages over any single LLMs. For self-refine, it is a few-shot method with human written 8 complex generate-refine examples, while the debate in our work is a zero-shot process, so it might be unfair to compare our work with self-refine. Also, we want to highlight that our work is complementary to self-consistency and self-refine, so in each debate round, we can apply self-consistency of self-refine to each LLM to produce a better argument and finally reach a better performance.
>
> ## ● Response 2 (Reasons to Reject 2)
> Take ChatGPT & Davinci-003 for example, our work produces an average of 3 arguments in each question, which means ChatGPT and Davinci-003 will forward 3 times in total for each question. While for self-consistency, it would forward 1 time but sample 40 arguments for each question, which is more costly. For self-refine, it should iterate 4 times for each question, so the LLM should forward 8 times (each iteration contains a feedback and a refine processes), which is also more costly than ours. Table 2 below show to detailed cost of self-consistency, self-refine, and FORD.
>
> | Method           | Average number of forwards for each question | Num of sampled arguments in each forward |
> | ---------------- | -------------------------------------------- | ---------------------------------------- |
> | Self-Consistency | 1                                            | 40                                       |
> | Self-Refine      | 8                                            | 1                                        |
> | FORD (Ours)      | 3                                            | 1                                        |
>
> ***Table 2: The cost of self-consistency, self-refine, and FORD***
>
> ## ● Response 3 (Questions)
> Plesase refer to Response 1 and Response 2.

---

### Official Review · Reviewer_LsCJ · 2023-08-04

**Soundness:** 3

**Excitement:**

3: Ambivalent: It has merits (e.g., it reports state-of-the-art results, the idea is nice), but there are key weaknesses (e.g., it describes incremental work), and it can significantly benefit from another round of revision. However, I won't object to accepting it if my co-reviewers champion it.

**Paper Topic And Main Contributions:**

The paper proposes the FORD procedure for commonsense QA datasets, in which three LLMs are paired together (proponent, opponent, and a third more minor judge). The models debate and potentially come to a consensus. The paper studies whether such a procedure can lead to improved answer quality, and whether the models are likely to agree/converge in their thinking. Experiments mostly show  the FORD method leads to minor gains when the proponent/opponent have similar base performance, but leads to lower performance when the model's base performance is mismatched. The paper finds that models tend to converge rapidly in most cases. A roundtable version experiment with 3 participants finds similarly that improving on the best single model performance (GPT4) is challenging.

**Questions For The Authors:**

- The FORD method relies on three LLMs: proponent, opponent and judge. In the paper, you clearly define the first two roles, but I could not understand which LLM plays the role of the judge. I could not find this information (although I could have missed it), and it is not defined in Section 3.3, or at any mention of "judge". You mention in 7.1 that when switching the judge to GPT4, you see a very minor performance improvement, but do not specify what the judge is in the first place. Seeing that the judge is producing the final label, it seems like a very important element. Is there evidence that the judge considers the content of the debate when making the final decision? Does it happen that the judge disregards the debate and makes its own decision? (this would explain the minor GPT4 gain of 7.1), but seems problematic.
- Why the need for the judge's summary? Is it used in the pipeline, or simply for bookkeeping?
- How is the debate halted? Can either model decide to halt the debate, or do you always go until the max number of turns?
- Any ideas on baselines that you could be compared to that could help prove that the debate aspect is important and helps improve answer quality?
- I am not sure I understand the concept of "inter-consistency". I understand that it is being contrasted to self-consistency, but can different models really be "consistent"? Shouldn't it be called "model agreement"? What is there to be "inter"-consistent about?

**Reasons To Accept:**

- Several ideas are innovative, including the INCON measure (to measure agreement level over time), and the entire FORD protocol, which involves the debate, the summarization & conclusion.
- The paper does show that in some cases, the FORD protocol does lead to minor improvements on the task.
- The paper involves broad experiments across 7 datasets, which is comprehensive, with results mostly holding across datasets.

**Reasons To Reject:**

- It seems like the conclusion is that in the end, using a better LLM (GPT4) is more important than FORD strategies that lead to only minor gains. The fact that mismatched debates (with a weaker LLM) leads to performance degradation is also highly problematic. It is often not known in advance whether two LLMs have similar performance on a task. Would one really risk a mismatched situation which could lead to a large drop for the potential benefit of a minor performance boost?
- The baselines (single model and then soft/hard majority voting) could be improved and enhanced. It seems like the FORD process resembles a fuzzy retrieval system, which extracts more context from an LLM. It would be valuable to compare to a retrieval-enabled LLM as a baseline. Does FORD elicit LLMs to change their reasoning, or is it simply a noisy retrieval process to retrieve relevant content before making a decision (in some sense, a multi-step chain-of-thought)? More baselines are needed to fully demonstrate the value of FORD.
- There is a lack of clarity on a few key concepts in the paper, which can be addressed (see questions to the authors).

**Reproducibility:**

3: Could reproduce the results with some difficulty. The settings of parameters are underspecified or subjectively determined; the training/evaluation data are not widely available.

**Reviewer Confidence:**

4: Quite sure. I tried to check the important points carefully. It's unlikely, though conceivable, that I missed something that should affect my ratings.

---

> ### Author Rebuttal · Authors · 2023-08-29
>
> Thanks for the valuable feedback and insightful comments!
> ## ● Response 1 (Reasons to Reject 1)
> First, we suppose better fair LLMs pair with FORD is more important instead of better single LLMs. In fair situation, there are absolute improvements with FORD (average 2.34% by ChatGPT & Davinci-003, average 4.12% by LLaMA & Vicuna). If we can find LLMs as strong as GPT-4, the performance can be also improved by FORD (Line 350). And we believe there will be LLMs catch up with GPT4, or, even GPT4 can have different instances to support the situation of fair debate.
>
> Second, to ensure the integrity of the research, we further investigate mismatched debates. We found it would lead to performance degradation for the stronger LLM while helping the weaker LLM to improve the performance.
>
> Third, our work is mainly to investigate in which situation the LLMs can collaborate better (Line 36), mismatched situation is not ideal according to the results (Table 3 in the paper) and fair situation is a more ideal situation (Table 2 in the paper).
>
> Therefore, the main idea of our work is to systematically investigate whether LLMs can collaborate towards a shared goal, which offers many insights and lays the foundation for developing future collaboration methods. Obtaining better performance is not our only pursuit. Debate among fair LLMs is better for collaboration, and better fair LLMs can lead to better performance with FORD. Besides, a stronger LLM (if available) can be introduced for further improvements with acceptable costs.
>
> ## ● Response 2 (Reasons to Reject 2)
> In some sense, our work is an expansion of CoT towards multi-step and multi-agent Chain-of-Thought (CoT). Generally speaking, previous works like self-consistency (Wang et al. 2023) can be seen as one kind of multi-step expansion. However, self-consistency is using the same LLM which usually produces very similar cot, leading to limited insights and improvements. FORD further introduce multiple LLM agents via a human-like debate process, encouraging LLMs to debate each other for diverse reasoning perspectives (refer to the case study in Sec 7.3). It is more like an interactive process than a retrieve-then-decide process. Moreover, we randomly select two datasets for the comparison with self-consistency. Table 1 below also demonstrates the advantage of our work over self-consistency. Of course, self-consistency (kind of breadth expansion) can be integrated into our framework (kind of depth expansion) to improve each individual LLM. We leave it in the future. Hence, our work can improve the scope of different steps of cot, resulting in new aspects and new ideas for LLM-based multi-agent collaboration to conduct reasoning.
>
> [Wang et al., 2023] Self-Consistency Improves Chain of Thought Reasoning in Language Models
>
> | Method                             | $\alpha$NLI | e-CARE |
> | ---------------------------------- | -------- | ------ |
> | Self-Consistency (ChatGPT)         | 77.84    | 82.47  |
> | Self-Consistency (Davinci-003)     | 79.89    | 76.44  |
> | FORD (ChatGPT & Davinci-003, Ours) | **81.35**    | **83.74**  |
>
> ***Table 1: the results of self-consistency and FORD on two datasets.***
>
> ## ● Response 3 (Question 1)
> In the main framework, the judge is a heuristic method, not an LLM (Line 214). The heuristic method is described as follows. For each response in the debate process, the proponent or opponent would give an updated answer and argument based on the previous arguments. Hence, if the proponent and opponent LLMs reach a consensus, we treat the agreed answer as the final label. When the LLMs do not reach an agreement, we assign equal weights to the answers of all the responses to get the final label. When switching to GPT-4 as the judge, take ChatGPT & Davinci-003 for example, only the debate processes of the inconsistent samples (19.6%) will be sent to GPT-4 to evaluate the debate processes and produce the final labels. It obtains minor gains because the number of inconsistent samples is quite small (around 300 samples in each dataset). Actually, the upper bound of the improvement when introducing GPT-4 as the judge is 7.27%, GPT-4 as the judge can improve by 9.17% of the upper bound.
>
> When switching to GPT-4 as the judge, in the instruction, we will ask GPT-4 to give a summary and final answer only based on the debate process (Line 422 or Prompt 15 in the Appendix).
>
> ## ● Response 4 (Question 2)
> Just like the human debates and for the completeness of the research, we adopt a judge in the pipeline to summarize the debate process and give a final answer. Ideally, we can adopt humans as the judge in the future, and obtain better decisions by designing a human-in-the-loop framework, which is an ongoing work.
>
> ## ● Response 5 (Question 3)
> When one LLM compromises with the other (refer to Response 3 above) or debate round reaches 6, the debate will halt, the halt can happen by any LLM. Take ChatGPT & Davinci-003 for example, the average number of debate round is 3 on all datasets. Therefore, LLMs will reach a consensus on most of the samples (refer to Figure 4 in the paper).
>
> ## ● Response 6 (Question 4)
> We provide 4 baselines (two single LLMs, two collaboration methods) as a comparison to analyze the importance of debate, quantitatively and qualitatively. Quantitatively, our work (FORD) could achieve better performance than the independent LLMs and simple LLMs collaboration methods (refer to Table 2 in the paper). Table 1 in Response 2 also shows the advantage of FORD over self-consistency. Qualitatively, FORD could provide evolutional and multi-aspect rationale or cot to improve the quality of interpretability and prediction, which can be seen in the example below or the case study (Sec 7.3).
>
> ````
> Question: The woman lost her place in line. What was the cause of this?
>
> Options: (A) More people entered the line. (B) She stepped out of the line.
>
> Golden Answer: (B)
>
> **ChatGPT:** The answer is (B). This answer is more plausible because it directly states that the woman herself did something that caused her to lose her place in line. Option (A) suggests that other people caused her to lose her place, which is possible but not necessarily the most likely scenario.
>
> **Davnci-003:** If the woman lost her place in line, it is more likely that it was due to more people entering the line, thus pushing her back. Therefore, answer (A) is more plausible.
>
> **ChatGPT:** While it is possible that the woman stepped out of the line, it is more likely that more people entered the line and pushed her back, causing her to lose her place. This is a common occurrence in lines, especially in crowded places. Therefore, the answer is (A).
>
> **GPT-4 Summary:** ChatGPT argues that the woman stepping out of the line is the more plausible cause of her losing her place, as it directly involves her action. Davinci-003, however, believes that the addition of more people in the line, pushing her back, is a more common occurrence and thus more likely. All arguments have merit, but (B) She stepped out of the line is more plausible as it directly involves the woman's actions leading to the loss of her place.
> ````
>
> ## ● Response 7 (Question 5)
> Self-consistency is for different answers produced by the same model, so we introduce inter-consistency for different answers of multiple models, or multiple instances of the same model (like ChatGPT0301 and ChatGPT0624). We agree that model agreement can also well explain the idea. We just follow the conventions and consider the case of multiple instances of the same model.

---

### Official Review · Reviewer_R6wy · 2023-08-06

**Typos Grammar Style And Presentation Improvements:** None.
**Soundness:** 3

**Excitement:**

4: Strong: This paper deepens the understanding of some phenomenon or lowers the barriers to an existing research direction.

**Justification For Ethical Concerns:**

None.

**Missing References:**

None.

**Paper Topic And Main Contributions:**

This paper investigated inter-inconsistency among LLMs when in collaboration. To achieve this, the authors proposed a debate framework to conduct debates among two or more LLMs, and they use this to examine whether LLMs can collaborate effectively to achieve a shared goal. There are three different scenarios in the experiments: fair debate, mismatched debate, and roundtable debate. In my understanding, there are two important contributions: this is the first work that systematically researches inter-consistency among LLMs, and findings from this paper will lay a kind of foundation for collaboration research in the future.

**Questions For The Authors:**

1. Please clarify a bit, in Fig.2, why there are four different predictions for the only one sample in step 1, and are there any explanations why step 1 and step 2 use different samples?

2. how do you get this claim, "It is mainly because GPT- 4 can assign higher weights to more convincing arguments, then draw more precise conclusions."?

**Reasons To Accept:**

1. This is the first work that systematically researches inter-consistency among LLMs, and the previous works are all for the self-inconsistency of single LLMs. This would be interesting to researchers who are going to do research on LLM collaboration in other tasks like text generation, although this work is based on commonsense reasoning.

2. The paper proposed a framework for examining inter-inconsistency among LLMs and provides interesting findings which can be used in research of LLM collaboration. They find that LLMs possess the spirit of collaboration to achieve a consensus for solving a shared
goal. Debates can improve the performance and inter-consistency of LLMs. Stronger LLMs can be distracted by weaker LLMs when the debates are mismatched.

3. The paper is well-written. The motivation, methodology, experiments and analysis are all clear and it is easy to understand.

**Reasons To Reject:**

Although the idea of the paper is interesting, I have several concerns:

1. There is inconsistency in writing. Line 417 says "investigate GPT-3 as the judge" but the whole section 7.1 uses GPT-4 as the judge. Not sure how this happened, which is the one authors used.

2. Details to get the debate summary are missing or inconsistent.
2.1 In the main text, it is when there is no consensus, the two stances will be assigned equal weights, while in the appendix, it is voting for the final answer.
2.2 in section 3.3, I do not think that one can use the small group of words, “while”, “defend”, “argue”, “compromise”, “claim”, “suppose”, and “agree” to construct the example summary without knowing whether the model compromises or refutes.

3. As most of the results are based on closed-source models, I'm afraid that the results are not easily reproducible even if the temperature is set to 0.0, as temperature 0.0 cannot guarantee to generate deterministic results.

**Reproducibility:**

2: Would be hard pressed to reproduce the results. The contribution depends on data that are simply not available outside the author's institution or consortium; not enough details are provided.

**Reviewer Confidence:**

4: Quite sure. I tried to check the important points carefully. It's unlikely, though conceivable, that I missed something that should affect my ratings.

---

> ### Author Rebuttal · Authors · 2023-08-29
>
> Thanks for the valuable feedback and insightful comments!
> ## ● Response 1 (Reasons to Reject 1)
> GPT-4 is the only judge throughout Sec 7.1. Line 417 is a typo. We will correct it in the revised version.
>
>
> ## ● Response 2 (Reasons to Reject 2)
>  When it's the turn of any LLM to debate, it will be asked to give an updated stance (answer) and a corresponding argument (explanation) based on all previous arguments. If the updated stance (answer) is the same as the **latest** stance (answer) of other LLMs, then they reach a consensus and the debate will halt, otherwise, the debate will go on. Take ChatGPT & Davinci-003 for example, previously Davinci-003 answers B, ChatGPT answers A, and Davinci-003 answers B. Then if ChatGPT answers B, then there is a consensus. There is no consensus if ChatGPT insists on stance A.
>
> (1) When there is no consensus, the answers (stances) produced by each round of the whole debate process will be used to vote for the final answer (kind of assigning equal weights). When there is a consensus, the final answer is the agreed answer (stance).
>
> (2) If the LLM gives the same answer as the other, then it compromises, otherwise, it refutes. Hence, we can use a group of words to construct a summary. In detail, we can use a heuristic method for summarization. "Claim" and "suppose" can be used to summarize the stance selection and explanation generation process (Step 1 in FORD). "Defend", "argue", and "while" can used when the model refutes. "Compromise" and "agree" can be used when the model compromises. We will add the necessary details in the revised version.
>
> ## ● Response 3 (Reasons to Reject 3)
>
> | Debate                                     | PIQA                                     | e-CARE                                   |
> | ------------------------------------------ | ---------------------------------------- | ---------------------------------------- |
> | ChatGPT & Davinci-003  (Fair)              | (**84.60**/85.53/85.42)  *85.18 ± 0.41* | (**83.74**/84.78/84.59)       *84.37 ± 0.45* |
> | ChatGPT & GPT-4 (Mismatched)               | (**92.71**/93.85/93.69)       *93.42 ± 0.50* | (**85.96**/86.33/86.29)       *86.19 ± 0.17* |
> | ChatGPT & Davinci-003 & GPT-4 (Roundtable) | (**86.02**/85.58/85.53)       *85.71 ± 0.22* | (**84.78**/84.21/84.17)       *84.39 ± 0.28* |
>
> ***Table 1: The multiple runs and *mean* results of each debate on each dataset. The bold result of each debate on each dataset is the score in our paper.***
>
> We randomly select two datasets and three debates (one debate for each of fair, mismatched, and roundtable situations) for three runs on each dataset. Table 1 above shows the results of each run and the mean and standard variation of the accuracy across multiple runs on each dataset. We can find that setting temperature to 0.0 can achieve a relatively stable performance. Using a temperature of 0.0 is a common method to improve reproducibility. It is a common question. Many previous works use a very low temperature for research (Sun et al., 2023; Liu et al., 2023).
>
> [Sun et al., 2023] Recitation-Augmentation Language Models
>
> [Liu et al., 2023] The Magic of IF: Investigating Causal Reasoning Abilities in Large Language Models of Code
>
> ## ● Response 4 (Question 1)
>
> There are multiple samples in Step 1, which are differentiated by colors. Only one sample’s content is shown in step1 due to space limitation. Therefore, in Step1, four different predictions are for four different samples, and only the samples (green and orange) without consensus are fed into Step2. We will make it clearer in the revised version.
>
> ## ● Response 5 (Question 5)
> This claim means that GPT4 can make better judgements by trusting more convincing arguments than other judges. This assumption is based on our experimental results and previous findings  (Liu et al., 2023). On the one hand, the results with GPT-4 as the judge are better than those without GPT-4 (Table 6 in the paper). Clearly, GPT-4 can make better judgements among different arguments and choose the convincing ones for final results. Likely, it assigns higher weights to correct arguments in voting (refer to the case below). On the other hand, previous works have proved that GPT-4 as an evaluator has a high agreement with human evaluators (Liu et al., 2023). This also supports our assumption that GPT-4 can make better judgements.
>
> [Liu et al., 2023] GPTEVAL: NLG Evaluation using GPT-4 with Better Human Alignment
>
> Below is a real debate of ChatGPT & Davinci-003, on which we use GPT-4 for summary, GPT-4 actually assigns a higher weight to ChatGPT’s first argument and draws the right prediction.
> ```
> Question: The woman lost her place in line. What was the cause of this?
>
> Options: (A) More people entered the line. (B) She stepped out of the line.
>
> Golden Answer: (B)
>
> ChatGPT: The answer is (B). This answer is more plausible because it directly states that the woman herself did something that caused her to lose her place in line. Option (A) suggests that other people caused her to lose her place, which is possible but not necessarily the most likely scenario.
>
> Davnci-003: If the woman lost her place in line, it is more likely that it was due to more people entering the line, thus pushing her back. Therefore, answer (A) is more plausible.
>
> ChatGPT: While it is possible that the woman stepped out of the line, it is more likely that more people entered the line and pushed her back, causing her to lose her place. This is a common occurrence in lines, especially in crowded places. Therefore, the answer is (A).
>
> GPT-4 Summary: ChatGPT argues that the woman stepping out of the line is the more plausible cause of her losing her place, as it directly involves her action. Davinci-003, however, believes that the addition of more people in the line, pushing her back, is a more common occurrence and thus more likely. All arguments have merit, but (B) She stepped out of the line is more plausible as it directly involves the woman's actions leading to the loss of her place.

---

### Meta-Review · Area_Chair_AHES · 2023-09-15

**Recommendation:** 4

**Metareview:**

This paper introduces the FORD framework, a novel approach to investigate the inter-consistency and collaboration capabilities of Large Language Models (LLMs) by facilitating debates among them. Through various scenarios like fair debate, mismatched debate, and a roundtable setup, the research assesses whether LLMs can effectively collaborate and converge in their thinking to improve answer quality, especially in commonsense QA datasets.

The strengths of the paper include:
1) This paper presents early research that systematically evaluates inter-consistency among LLMs
2) Several ideas are very innovative such as INCON measure.
3) The paper conducted a comprehensive experiment across 7 datasets.
4) the paper is well written and easy to read

The main concerns from reviewers are:
1) some parts of the paper require clarification
2) Most of the results rely on closed-source models, which affect the reproducibility.
3)  Some conclusions are inconsistent.
4) experimental comparison is inadequate.

---

### Decision · Program_Chairs · 2023-10-07

**Decision:**

Accept-Findings

**Comment:**

This paper introduces the FORD framework, a novel approach to investigate the inter-consistency and collaboration capabilities of Large Language Models (LLMs) by facilitating debates among them. Through various scenarios like fair debate, mismatched debate, and a roundtable setup, the research assesses whether LLMs can effectively collaborate and converge in their thinking to improve answer quality, especially in commonsense QA datasets.

The strengths of the paper include:
1) This paper presents early research that systematically evaluates inter-consistency among LLMs
2) Several ideas are very innovative such as INCON measure.
3) The paper conducted a comprehensive experiment across 7 datasets.
4) the paper is well written and easy to read

The main concerns from reviewers are:
1) some parts of the paper require clarification
2) Most of the results rely on closed-source models, which affect the reproducibility.
3)  Some conclusions are inconsistent.
4) experimental comparison is inadequate.